# On the Cutting Edge of Oral Cancer Prevention: Finding Risk-Predictive Markers in Precancerous Lesions by Longitudinal Studies

**DOI:** 10.3390/cells11061033

**Published:** 2022-03-18

**Authors:** Madeleine Crawford, Eliza H. Johnson, Kelly Y. P. Liu, Catherine Poh, Robert Y. L. Tsai

**Affiliations:** 1Institute of Biosciences and Technology, Texas A&M Health Science Center, Houston, TX 77030, USA; madeleinevcrawford@outlook.com (M.C.); elizahjohnson@email.arizona.edu (E.H.J.); 2Department of Translational Medical Sciences, Texas A&M Health Science Center, Houston, TX 77030, USA; 3Faculty of Dentistry, The University of British Columbia, Vancouver, BC V6T 1Z3, Canada; keliu@bccrc.ca

**Keywords:** DNA methylation, epigenetic, genetic mutation, long-term follow-up, risk assessment, translational study

## Abstract

Early identification and management of precancerous lesions at high risk of developing cancers is the most effective and economical way to reduce the incidence, mortality, and morbidity of cancers as well as minimizing treatment-related complications, including pain, impaired functions, and disfiguration. Reliable cancer-risk-predictive markers play an important role in enabling evidence-based decision making as well as providing mechanistic insight into the malignant conversion of precancerous lesions. The focus of this article is to review updates on markers that may predict the risk of oral premalignant lesions (OPLs) in developing into oral squamous cell carcinomas (OSCCs), which can logically be discovered only by prospective or retrospective longitudinal studies that analyze pre-progression OPL samples with long-term follow-up outcomes. These risk-predictive markers are different from those that prognosticate the survival outcome of cancers after they have been diagnosed and treated, or those that differentiate between different lesion types and stages. Up-to-date knowledge on cancer-risk-predictive markers discovered by longitudinally followed studies will be reviewed. The goal of this endeavor is to use this information as a starting point to address some key challenges limiting our progress in this area in the hope of achieving effective translation of research discoveries into new clinical interventions.

## 1. Introduction

Oral cancer is the 18th (out of 36) most common cancer worldwide, with an annual incidence of 377,713 and a mortality of 177,757 in 2020 [1], and is the 8th and 15th most common cancer for males and females in the US, respectively [2]. Its five-year survival rate remains at 66% (American Cancer Society, 2021) (https://www.cancer.org/cacer/oral-cavity-and-oropharyngeal-cancer/detection-diagnosis-staging/survival-rates.html; accessed on 8 March 2022). The etiology of oral squamous cell carcinoma (OSCC) can be categorized into three major groups, including: (1) oral habits associated with tobacco, heavy alcohol, and betel nut chewing; (2) human papilloma virus (HPV) infection; and (3) no known risk factor. In the US, tobacco and heavy alcohol usage remain by far the most common etiological factor, whereas betel nut chewing is more common in Southeast Asia. HPV is a rare cause for OSCC. HPV-negative non-smokers represent a small subset of OSCC patients that are relatively overrepresented by females [3,4,5,6]. Tobacco-related OSCC occurs most commonly on the tongue and the floor of the mouth. It is believed to develop through a premalignant stage of epithelial dysplasia. Dysplastic changes of oral keratinocytes start in the basal and parabasal cell layers, showing hyperchromatism, pleomorphism, increased nuclear-to-cytoplasmic ratio, large and prominent nucleoli, increased mitotic activity, abnormal mitotic figures, and altered epithelial architecture and maturation pattern. Oral epithelial dysplasia is classified either as low-grade, including mild and moderate dysplasia, when cytomorphological changes are confined to the lower half of the epithelium, or as high-grade (severe dysplasia) when changes involve more than half of the epithelial thickness according to the 2017 WHO criteria [7]. This classification was recently challenged for its ability to predict risks, and, as a result, a two-tier grading system was proposed [8]. However, considering the subjectivity in grading and, hence, inter- as well as intra-observer discrepancies, its predictive value requires further validation. This highlights the need for more objective markers for risk prediction of malignant transformation.

## 2. Risk-Predictive Markers Based on a Longitudinally Followed Study Design

Biomarkers are biological identifiers that can provide crucial information on disease development, diagnosis, and progression. A reliable biomarker needs to demonstrate a high sensitivity and specificity in its power of prediction or differentiation [9]. The terminology used to describe biomarkers serving various clinical purposes can be confusing. Most researched biomarkers fall within three categories: diagnostic, predictive, and prognostic. Diagnostic markers refer to those that differentiate different lesion types or stages without follow-up data (Figure 1A). They are by far the most common type of biomarkers, discovered by comparing cross-sectional samples collected from different patients [10,11,12]. Predictive markers refer to those that indicate the risk of a disease (e.g., cancer) without intervention (Figure 1B). They are discovered by comparing pre-disease/cancer samples with follow-up data indicating their outcome of disease/cancer development. Prognostic markers refer to those that forecast the outcome of a disease/cancer (Figure 1C). They are discovered by comparing disease/cancer samples, after being diagnosed, with follow-up data indicating their outcome with/without intervention. Although diagnostic, predictive, and prognostic markers each serve a particular purpose, occasionally distinction between them can prove difficult. Some diagnostic markers may also play a role in predicting the cancer risk of precancerous lesions or vice versa.

Current pathological grading of OPLs provides information on the severity of dysplasia that correlates with risk of cancer development to some degree and remains the gold standard in predicting risk of oral cancer in the clinic. High-grade dysplasia has a high (35%) and better predictive value of cancer and, therefore, is recommended for treatment. On the other hand, low-grade dysplasia, which makes up the majority of the OPLs, has a low risk (4–11%) and poor predictive value [13] and is, therefore, not recommended for indiscriminate treatment, given its relatively low transformation rate and most frequent treatment complications, particularly for lesions that are diffuse or large in size. It is, therefore, clinically important to find predictive markers capable of stratifying high-risk vs. low-risk low-grade dysplastic lesions. To date, studies on this topic are quite limited due to the rarity of prospectively collected samples or retrospectively collected pre-progression samples with long-term follow-up data. This article will review published data on cancer-risk-predictive markers for OPLs identified solely by longitudinally designed studies, discuss the limitations and challenges related to the discovery and implementation of those markers, and muse over potential research trends in the future.

## 3. Molecular Markers

Molecular markers provide additional evidence to assess the cancer risk of OPLs. Based on the target of detection, they can be divided into protein, genetic, and epigenetic markers. Protein markers are frequently used in scientific research to identify specific antigens amongst mixed cell populations based on immunohistochemistry (IHC). Due to the complexity of molecular events involved in carcinogenesis, especially those occurring early in the disease process, recent studies have turned to genetic/epigenetic changes for insight into disease progression and prognosis. Genetic markers come in many forms, including single-nucleotide polymorphisms (SNPs), insertions, deletions, mutations, and duplications, which may be connected to the regulation of gene expression and function. RNA characteristics (e.g., transcriptome) also provide information on gene expression and may serve as potential markers [14]. More recently, epigenetic changes, including DNA methylation, histone modifications, and miRNA expression changes, have been investigated as potential markers in a variety of cancers. DNA methylation on promoter regions is the most commonly observed epigenetic change in carcinogenesis [15]. As DNA methylation signatures can be detected from a variety of body fluids, the use of epigenetic markers in a clinical setting is becoming ever more appealing.

## 4. Predicting the Cancer Risk of OPLs by Quantitative Pathology

The accumulation of genomic instability over time leads to phenotypic changes that can be used to differentiate malignant cells from their normal counterparts. Some of these changes are apparent enough to be detected at the routine (H&E) histopathological level. Under the premise that tumor cells need to acquire sufficient genetic changes to survive and aggressively progress, a subset of at-risk pre-malignant cells may also be phenotypically distinct, and their presence may predict the risk of cancer progression. The emergence of quantitative tissue pathology (QTP) has allowed the profiling of a plethora of microscopic characteristics at the single cell and subcellular level that cannot be gleaned directly by human eyes. These phenotypes could reflect the etiology of cancer and/or the consequence of its underlying pathogenic mechanism [16,17]. Guillaud et al. proposed a promising risk assessment tool, i.e., the nuclear phenotypic score (NPS), for oral cancer progression by building a phenotypic model to recognize nuclear phenotypes, such as nuclear size and shape and DNA amount and distribution, which are significantly discriminative between nuclei of normal, mild, moderate, severe dysplasia, carcinoma in situ, and SCC [18]. Most importantly, based on the best two-group cutoff, NPS was able to separate progressive (high NPS) lesions from non-progressive (low NPS) lesions. Overall, 71% of the high-NPS lesions advanced to cancer within 5 years as comparted to only 22% of the low-NPS lesions, a 10-fold increase in relative risk of progression to cancer based on the NPS level.

## 5. Cancer-Risk-Predictive Protein Markers for OPLs

Over the last few decades, IHC-based markers have been explored and applied exponentially in many human diseases. Between 1985 and 2006, publications pertaining to IHC markers have increased by over 10,000% [19]. The search terms used to identify studies on IHC-based oral cancer-risk-predictive markers include two core inclusion criteria: (1) primary samples from pre-malignant dysplasia and/or leukoplakia of any grade (low-grade, high-grade, mild, moderate, or severe) collected before they either advanced to cancer (progressive OPLs) or remained OPLs (static OPLs); and (2) samples with longitudinally followed outcomes. Additional terms used for selecting IHC-based studies were risk-predictive, OSCC, predictive biomarkers, pre-malignant oral lesions, immunohistochemistry predictive biomarkers, IHC OSCC risk factors, IHC oral dysplasia, IHC OSCC risk, and/or IHC OSCC prediction. Based on these search criteria, 19 studies were identified, from which 29 IHC-based markers were reported to have the potential of distinguishing progressive vs. static OPLs (Table 1).

### 5.1. Stem Cell Self-Renewal Factors

Stem cells and cancer cells share many common features, such as self-renewal and undifferentiated potential. Therefore, it should come as no surprise that the most researched protein markers for predicting the development of precancerous lesions into cancers are frequently also implicated in stem cell self-renewal [20]. A study by Zhang et al. reported a higher expression level of β-catenin, a core component of the WNT pathway that is critical for stem cell self-renewal, in OSCC-transformed oral leukoplakia compared to non-transformed oral leukoplakia with a median follow-up of 11.3 years in a univariate analysis (hazard ratio = 4.228, *p* = 0.001) [21]. The same study also identified cyclooxygenase 2 (COX2), c-Met (also called tyrosine-protein kinase Met or hepatocyte growth factor receptor HGFR), carbonic anhydrase 9 (CA9), Podoplanin (PDPN, a transmembrane glycoprotein associated with lymph node metastasis and poor survival in HNSCC), Ki-67, p16, p53, IMP3 (IMP U3 small nucleolar ribonucleoprotein 3), and c-Jun as potential risk-predictive markers. The WNT pathway also regulates SNAI1 (snail family transcriptional repressor 1) and AXIN2 (axin 2). One study showed that both SNAI1 and AXIN2 were expressed at higher levels in progressive (40.7% for SNAI1 and 26.3% for AXIN2) compared to static leukoplakia (8.7% for SNAI1 and 12.6% for AXIN2), and that both SNAI1 and AXIN2 were independent risk factors for transformation by multivariate analysis [22]. SMAD4 (SMAD family member 4) is a downstream target of the BMP (bone morphogenetic protein) pathway that cross-talks with the WNT pathway [23]. One study reported that low SMAD4 expression in oral leukoplakia is associated with increased malignant transformation and lymphocyte infiltration, suggesting that the combination of low SMAD4 expression and high lymphocyte infiltration may predict the risk of malignant transformation [24]. Notch1 is a cancer stem cell (CSC) marker and a signaling pathway necessary for tissue development and tumor progression. One study reported that oral leukoplakia that progressed to OSCC in five years showed a decrease in nuclear Notch1 expression and an increase in membranous Notch1 expression compared to those that remained static (*p* = 0.001), and that 38% of patients with membranous expression of Notch1 progressed to OSSC, compared to 13% of those without [25].

The ability to maintain genome integrity throughout DNA replication is an essential feature of self-renewing stem and cancer cells [26]. A 2016 study showed that ATM (ATM serine/threonine kinase) expression was found in 77.8% of progressive dysplasia and 49.4% of static dysplasia, and that yH2AFX expression was observed more in progressive OPLs (55.6%) than in static OPLs (23.5%) [27]. A recent study from our group examined the expression patterns of a stem and cancer cell self-renewal factor, nucleostemin (NS), in human oral dysplastic samples with longitudinally followed outcomes. Nucleostemin is a nucleolar GTP-binding protein that is highly expressed in stem and cancer cells belonging to a novel class of nucleolar GTPases [28,29]. NS plays a crucial role in self-renewal maintenance by promoting the repair of replication-induced DNA damage [30,31,32,33,34,35]. Our results revealed that cells with prominent nucleolar NS signals were more abundant in low-grade dysplasia that advanced to OSCC in 2–3 years compared to those remaining static for 7–14 years, suggesting that NS upregulation may be an early event in the progression of low-grade dysplasia to OSCC [36].

SOX2 (SRY-box transcription factor 2) has been implicated in maintaining CSC proliferation in head and neck SCC (HNSCC) [37]. One study showed that the OSCC progression rate in a five-year or longer follow-up period is 44% in patients with positive SOX2 expression and 13% in patients lacking SOX2 expression (*p* = 0.01) [38]. ALDH1 has also been proposed as a CSC marker for HNSCC. Positive expression of ALDH1 was found at a higher rate (73%) in progressive lesions of oral leukoplakia, either low-grade or high-grade dysplasia, compared to non-progressive lesions (50%) [39]. This study also showed that 58% of oral leukoplakia with positive PDPN progressed to OSCC, compared to only 23% of PDPN negative lesions (*p* = 0.010). One study reported that oral dysplastic lesions, either low-grade or high-grade, with positive NANOG (Nanog homeobox) expression in the nucleus or cytoplasm showed an increased risk of progression in five years compared to NANOG-negative lesions, and that NANOG expression correlated with the increase in dysplasia grade [40].

### 5.2. Tumor Suppressors

p53 is a tumor suppressor that plays a master role in determining the outcome of cells (DNA damage repair, cell cycle arrest, or apoptosis) in response to genomic damage. One study indicated that the peak of p53 expression may occur near or during the transition from OPLs to OSCC [41]. A study by Cruz et al. showed that 86% of dysplasias presenting with suprabasal p53 expression (regardless of the grade of dysplasia) progressed to OSCC, as compared to only 22% of dysplasias with negative p53 expression (*p* = 0.002) [42]. A later study by the same group reported the sensitivity (33%), specificity (83%), positive predictive value (67%), and negative predictive value (56%) of suprabasal p53 expression in predicting the cancer risk of OPLs (i.e., dysplasia and/or leukoplakia) [43]. Loss of another tumor suppressor, p16, was found in both progressive and non-progressive oral leukoplakia, but a significant association with p16 loss was observed only in progressive lesions (*p* = 0.013) [44].

### 5.3. Others

MAGE-A (MAGE family member A) proteins are known to be expressed in malignant lesions but not in normal tissue [45]. One study reported a higher MAGE-A expression in oral leukoplakia undergoing malignant transformation in five years compared to those remaining static, with a sensitivity of 85.4% and a specificity of 100% [46]. Another study also showed higher MAGE-A expression in progressive OPLs compared to non-progressive OPLs within a five-year follow-up window, with a positive predictive value of 93% and a negative predictive value of 74.3% [47]. A study by Wu et al. (2018) showed a higher risk of malignant transformation in oral dysplastic lesions with low transglutaminase 3 (TGM3) expression compared to those with high TGM3 expression [48]. One study examined S100A7 overexpression in oral leukoplakia and found 92.3% of progressive OPLs had elevated expression, compared to only 71.8% of non-progressive OPLs (*p* = 0.014) [49]. A study compared the expression of cortactin and FAK (protein tyrosine kinase 2) in oral dysplasia with a minimum follow up of five years or until malignant transformation and found that those lesions expressing high levels of both proteins displayed the highest incidence of OSCC, followed by those expressing a high level of one of the two proteins and last by those expressing both proteins at low-to-moderate levels [50]. One study utilized a genome-wide expression profile (Bonferroni method) to identify oral leukoplakia with known outcomes collected from a chemoprevention trial [51]. Overexpression of a tyrosine kinase receptor, MET, but not age, treatment arm, or histology, was the only independent predictive factor, showing a hazard ratio of 3.84 (*p* = 0.003) by multivariate analysis. 

**Table 1 cells-11-01033-t001:** Cancer-risk-predictive protein markers for OPLs reported by longitudinally designed studies.

Reference	Biomarker	Conclusions	Functions	Tissue	F/U (Years)	Strength
Zhang et al. [21]	COX-2, c-Met, β-catenin, CA9, PDPN, Ki-67, p16, p53, IMP3, c-Jun	Expression of all markers potentially risk-predictive. Significant differences in positive expression between groups was observed.	Stem Cell Self-Renewal	Oral Leukoplakia (T/N)	11.3 (median)	3.04–29.00 (HR)
Zhang et al. (2017) [22]	Axin2, Snail	Elevated expression of Snail and Axin2 significantly correlate to risk of malignant transformation.	Stem Cell Self-Renewal	Oral Leukoplakia (T/N)	10.8 (median)	4.41, 7.47 (HR)
Sakata et al. [24]	SMAD4	Low expression combined with elevated lymphocyte infiltration indicative of malignant risk.	Stem Cell Self-Renewal	Oral Leukoplakia (T/N)	Unknown	2.63 (HR)
Ding et al. [25]	Notch1	Expression significantly associated with dysplasia severity and OSCC development.	Stem Cell Self-Renewal	Oral Leukoplakia (T/N)	6.18 (median)	3.4 (HR)
Crawford et al. [36]	Nucleostemin	NS upregulation may be an early event in malignant transformation of low-grade dysplasia.	Stem Cell Self-Renewal	Oral Dysplasia (P/NP)	2–3 (NP), 7–14 (P)	*p* = 0.02–0.05
de Vicente et al. [38]	SOX2	SOX2 is an independent predictor of cancer risk in OL.	Stem Cell Self-Renewal	Oral Leukoplakia (T/N)	6.25 (median)	3.0–5.83 (HR)
Habiba et al. [39]	ALDH1, PDPN	Both markers can be used for determining risk of malignant transformation in OL.	Stem Cell Self-Renewal	LG & HG Oral Dysplasia (T/N)	2.08 (median)	2.91–3.64 (HR)
de Vicente et al. [40]	NANOG	Positive NANOG expression associated with progression to oral cancer-positive expression of both markers demonstrated higher risk.	Stem Cell Self-Renewal	LG & HG Oral Dysplasia (T/N)	5.08 (median)	2.01 (HR)
Cruz et al. (1998) [42]	p53	p53 expression pattern has prognostic potential for pre-malignant lesions.	Tumor Suppressor	LG & HG Oral Dysplasia (T/N)	3 (median)	*p* = 0.002
Cruz et al. (2002) [43]	P53	Suprabasal p53 expression is indicative of malignant transformation.	Tumor Suppressor	PMOL (T/N)	5.0 (mean)	29–33% Sensitivity,83–100% Specificity
Wu et al. [44]	p16	p16 may predict malignant transformation of OL.	Tumor Suppressor	Oral Leukoplakia (T/N)	Unknown	3.54 (OR)
Baran et al. [47]	MAGE-A	MAGE-A expression can be a reliable predictor of malignant transformation in progressing leukoplakia.	Melanoma Associated Antigen	Oral & Laryngeal Leukoplakia (T/N)	5	96.5% Specificity, 58.2% Sensitivity
Ries et al. [46]	MAGE-A	Positive expression in oral leukoplakia is significantly correlated to malignant transformation.	Melanoma Associated Antigen	Oral Leukoplakia (T/N)	5	*p* = 0.0001
Wu et al. [48]	TGM3	Suggests TGM3 takes part in malignant transformation and may predict progression.	Tumor Suppressor	Oral Leukoplakia (T/N)	4.75 (T), 7.92 (N) (median)	5.55 (HR)
Kaur et al. [49]	S100A7	Overexpression demonstrates association with risk of transformation, with cytoplasmic overexpression being most significant.	Cell Cycle & Differentiation	Oral Leukoplakia (T/N)	3.04 (median)	2.36 (HR)
de Vicente et al. [50]	Cortactin, FAK	Pre-malignant oral lesions with co-expression of both markers demonstrate high risk of OSCC development.	Tumor Progression & Metastasis	Oral dysplasia- leukoplakia, erythroplakia (T/N)	5 (minimum)	6.30 (HR)
Saintigny et al. [51]	MET	Overexpression in oral leukoplakia was associated with malignant transformation.	Cell Proliferation	Oral Leukoplakia (T/N)	6.08 (median)	3.84 (HR)
Weber et al. [52]	CD68, CD163	Elevated CD68 and CD163 significantly associated with malignant transformation. Suggests the value of macrophages as potential predictive markers.	Macrophage Infiltration	Oral dysplasia- mild, moderate, severe (T/N)	5 (full)	55.6–72% Sensitivity, 72.7–73.5% Specificity
Ries et al. [53]	PD1, PDL1	Overexpression of both markers may be indicative of cancer risk.	Cell Proliferation	Oral Leukoplakia (T/N)	5 (minimum)	50–76.5% Sensitivity, 72.3–93.6% Specificity

Abbreviations: Follow-Up (F/U), Transformed (T), Non-Transformed (N), Progressing (P), Non-Progressing (NP), hazard ratio (HR), odds ratio (OR).

Finally, a 2020 paper by Weber et al. investigated the possibility of differentiating progressive vs. non-progressive oral leukoplakia based on their tumor immune responses, i.e., macrophage infiltration and polarization [52]. They found that epithelial and subepithelial infiltration of CD68+ and C11+ macrophages were significantly higher in progressive oral leukoplakia compared to non-progressive lesions within the five-year follow-up period. The epithelial density of CD163+ cells was also higher in the progressive than the non-progressive lesions. Another study examined the expression of the immune checkpoint proteins, PD1 (programmed cell death protein 1) and PD-L1 (programmed death-ligand 1) [53]. Oral leukoplakias that transformed in five years showed differences in expression compared to non-transformed leukoplakias, where overexpression of both makers was indicative of malignant transformation. PD1 was significantly overexpressed in both epithelium (*p* = 0.0001) and sub-epithelium (*p* = 0.002) in transformed lesions compared to non-transformed lesions. On the other hand, PD-L1 epithelial overexpression nearly reached statistical significance (*p* = 0.06) and showed a sensitivity of 50% and a specificity of 93.6% in its correlation with progressive lesions. 

Despite their wide use in cancer diagnosis and therapeutic practices to provide valuable information on disease progression and prognosis, IHC markers suffer some limitations. IHC staining is subject to a variety of technical variations pertaining to sample acquisition, fixation, processing, preservation, antigen retrieval, and staining procedures. In addition, the significance of the IHC readout is subject to potential interpreter-dependent biases as there is no uniform standard in defining positive vs. negative signals. Finally, most IHC interpretations are qualitative by nature, prone to the subjectivity of the individual analyzing the samples [54].

## 6. Cancer-Risk-Predictive Genetic Markers for OPLs

Besides the same two core inclusion criteria as described for IHC markers, additional search terms used for selecting studies on genetic markers were risk-predictive, OSCC, predictive biomarkers, pre-malignant oral lesions, genetic predictive biomarkers, genetic OSCC risk factors, genetic OSCC risk, genetic biomarkers oral dysplasia, Loss of heterozygosity (LOH) in oral dysplasia, LOH OSCC prediction, and/or LOH OSCC risk. Based on these search criteria, four studies were identified exploring LOH in similar regions as genetic markers with the potential of distinguishing progressive vs. non-progressive OPLs (Table 2).

Among the many genetic mechanisms that may serve as biomarkers, few have been investigated for their risk-predictive potential. LOH is one that has been frequently studied for its role in malignant transformation of epithelial dysplasia and the development of OSCC. Multiple studies were identified that focused on the risk-predictive potential of LOH. An early 1996 study found that LOH at either 3p and 9p or both was identified in 51% of patients with OPLs and 37% of the 51% patients eventually developed oral cancers, suggesting that LOH in those regions might be early events in tumorigenesis [55]. Another study by Rosin et al. investigated genetic changes between progressive and non-progressive OPLs and determined LOH at regions 3p and 9p to be a necessary feature of progression, as nearly all progressive OPLs harbored this loss. It is worth noting that samples with losses in other regions (4q, 8p, 11q, and 17p) in addition to 3p and 9p also demonstrated a 33-fold increase in cancer risk [56]. One study used a prospectively recruited cohort of low-grade oral dysplasia and confirmed that lesions with LOH in the 3p/9p regions had a 22.6-fold higher risk of malignant transformation compared to lesions with 3p and 9p retention, and that the risk-predictive potential was further increased when combined with LOH at other sites (4q, 17p) [57]. There is also evidence suggesting that a combinatorial approach may increase the cancer-predictive power of LOH by including parameters such as histological changes, chromosomal polysomy, and p53 expression [58]. A later study by Graveland et al. found that lesions with both the 9p LOH and the p53 mutation showed a higher risk of transformation than lesions with 9p LOH alone [59]. Finally, from TCGA molecular profiles of OSCC tumors, OSCC exhibited mutations in tumor suppressor genes at the same loci targeted by LOH, including *CDKN2A* (cyclin-dependent kinase inhibitor 2A) at 9p21 and *TP53*. Based on our search criteria, none of these reported genetic biomarkers have been validated for their progressive risk by a longitudinal study design.

**Table 2 cells-11-01033-t002:** Cancer-risk-predictive genetic markers for OPLs reported by longitudinally designed studies.

References	Biomarker	Conclusions	Tissue	F/U (Years)	Strength
Mao et al. [55]	LOH at 3p, 9p	Losses in these regions are frequent early genetic events in OPLs. Cancer developed more quickly in groups with LOH in regions 3p and/or 9p than those without LOH.	Oral Leukoplakia (T/N)	5.25 (median)	*p* = 0.039
Rosin et al. [56]	LOH at 3p, 9p, 4q, 8p, 11q, 17p	LOH at 3p and/or 9p exhibit increased risk of cancer development. Risk significantly increased in patients with losses on additional regions.	Hyperplasia, mild and moderate oral dysplasia (P/NP)	0.5 (minimum)	3.75, 33.4 (RR)
Zhang et al. (2012) [57]	LOH at 3p, 9p, 4q, 17p	LOH at 3p and/or 9p indicates risk for malignant transformation. Risk further increases when combined with LOH at additional sites.	Oral Dysplasia (P/NP)	3.7 and 3.6 (median)	22.6 (HR)
Graveland et al. [59]	LOH at 9p and p53 mutation	TP53 mutation correlated with losses at 17p and 9p. Losses at 9p significantly associated with risk of transformation.	Oral Leukoplakia (P/NP)	1.5 (median)	*p* = 0.014

Abbreviations: Follow-Up (F/U), Transformed (T), Non-Transformed (N), Progressing (P), Non-Progressing (NP), hazard ratio (HR), relative risk (RR).

## 7. Cancer-Risk-Predictive Epigenetic Markers for OPLs

Epigenetic regulation has emerged as a mechanism of intense research interest that captures the early impact of environmental insults on the genome and may provide key information on the progression of complex diseases, such as cancers, metabolic disorders, cardiovascular diseases, and neurodegenerative diseases [60]. Besides the two core inclusion criteria, additional search terms used for identifying studies on epigenetic risk-predictive markers included risk-predictive, OSCC, predictive biomarkers, pre-malignant oral lesions, epigenetic predictive biomarkers, epigenetic OSCC risk factors, epigenetic OSCC risk, epigenetic biomarkers oral dysplasia, methylation in oral dysplasia, methylation OSCC prediction, methylation OSCC risk, miRNA OSCC prediction, miRNA OSCC risk, and/or histone modification OSCC. Nine studies were identified from these searches (Table 3). 

### 7.1. DNA Methylation

Aberrant methylation has been observed as an early molecular event in oral carcinogenesis and, therefore, may serve as a risk-assessment marker for cancer prediction and prevention [61,62]. Most published studies on DNA methylation in oral carcinogenesis focused on analyzing samples of different lesion types collected from different patients (cross-sectional), which provide little predictive value in evaluating the future outcome of OPLs. Some studies analyzed dysplastic samples with different follow-up outcomes but used gene-specific approaches, which did not allow for the discovery of new targets.

The gene-specific approaches used to identify potential risk-predictive markers in OPLs most frequently involve promoter methylation of tumor suppressor genes. Inactivation of p16^INK4a^ via CpG methylation has been described as a key event in epithelial dysplasia, with multiple longitudinal-model studies suggesting its role in malignant conversion and risk prediction. One study found increased hypermethylation of p16^INK4a^ (57.7%) and p14^ARF^ (3.8%) as well as mutations and deletions in those genes in oral dysplastic lesions [63]. p16^INK4a^ hypermethylation was also associated with LOH on two or more of the following three markers: IFNα, D9S1748, and D9S171. However, follow-up information was not yet available. Therefore, the significance of those DNA methylation changes in predicting cancer progression remains to be determined. Four studies explored the DNA methylation patterns of p16^INK4a^ in longitudinally followed oral dysplasia and observed increased DNA methylation of p16^INK4a^ in progressive compared to non-progressive dysplasia [64,65,66,67]. The following three studies utilized the same patient cohort for prospective study. Cao et al. identified an association between p16^INK4a^ hypermethylation and malignant transformation with 63.6% sensitivity and 67.9% specificity [67], whereas Liu et al. found an association with 62% sensitivity and 76% specificity [65,66]. Together, these studies suggest that p16 methylation is a frequent event preceding cancer development in OPLs and serves as a cancer-risk-predictive marker for OPLs. 

To date, only seven published studies have investigated oral carcinogenesis (not HNSCC) using genome-wide approaches [68,69,70,71,72,73,74]. All seven studies used microarray-based platforms (e.g., Infinium Human Methylation 27K or 450K BeadChip or Agilent 4 × 44 CGH Microarray), which, unlike the whole-genome bisulfite sequencing (WGBS) approach, survey only a small percentage (up to ~2%) of the total epigenome. Five of the seven studies were based on a cross-sectional design, with three looking for DNA methylation differences between paired OSCCs and their adjacent normal tissues [68,69,70] and two looking for differences between non-paired OSCCs and normal tissue [71,74]. One of the seven studies analyzed the TCGA methylomic data using the Infinium Human Methylation 450 BeadChip to determine the DNA methylation differences between OSCCs with different survival outcomes (i.e., prognostic markers) [73]. Only one of the seven studies provided a genome-wide analysis of DNA methylation changes between 12 pairs of oral dysplastic samples with different follow-up outcomes, with progressive samples developing OSCCs in 2.15 years (mean) and non-progressive samples remaining dysplasias for 7.64 years (mean) [72]. Using the Agilent 4 × 44 CGH Microarray, which contained 44,674 probes that covered 8369 genes, Foy et al. identified hypermethylation on 86 genes and hypomethylation on Long Interspersed Elements 1 (LINE1) in patients who developed OSCC compared to those who did not. LINE1 is distributed widely across the genome, and its DNA methylation status can be used to indicate global DNA methylation. Most of the 86 hypermethylated genes were also found to be downregulated and their promoter regions were hypermethylated in OSCCs compared to normal tissue. Hypermethylation on the promoter regions of angiotensin II receptor type 1 (AGTR1), forkhead box I2 (FOXI2), and proenkephalin (PENK) was further validated by pyrosequencing (*p* = 0.003). Overall, oral carcinogenesis appears to be associated with global hypomethylation on CpG islands and hypermethylation on the promoter regions of specific genes that presumably play a tumor suppression role.

**Table 3 cells-11-01033-t003:** Cancer-risk-predictive epigenetic markers for OPLs reported by longitudinally designed studies.

Reference	Biomarker	Conclusions	Tissue	F/U (Years)	Strength
Hall et al. [64]	p16 Promoter Methylation	Presence of Promoter methylation of p16 is a potential predictor of malignant transformation.	Oral Leukoplakia, erythroplakia (T/N)	3 (minimum)	*p* = 0.002
Cao et al. [67]	p16 Methylation	Higher rate of progression to cancer in patients with positive p16 methylation.	Mild/Moderate Oral Dysplasia (T/N)	3.82 (median)	3.7 (OR)
Liu et al. (2015) [65]	p16 Hypermethylation	Positive p16 methylation significantly increased in transformed cases. Presents p16 Methylation as a definitive marker for determining malignant risk.	Mild/Moderate Oral Dysplasia (T/N)	3.42 (median)	*p* = 0.006
Liu et al. (2018) [66]	p16 Methylation	Progression to malignant transformation was significantly increased for patients with positive p16 methylation.	Oral leukoplakia, lichen planus, or discoid lupus erythematosus (T/N)	3.42 (median)	2.67 (OR)
Foy et al. [72]	AGTR1, FOXI2, PENK Promoter Methylation; LINE1 Hypomethylation	Patients with a high methylation index experienced worse Oral-Cancer-Free Survival. CpG Island Methylation may be an early event in OSCC.	Oral Premalignant Lesions (Unspecified) (T/N)	7.64 (N), 2.15 (T) (median)	*p* = 0.003
Philipone et al. [75]	miRNAs: 208b-3p, 204-5p, 129-2-3p and 3065-5p	Expression of the indicated miRNAs along with age and histology proved to be indicative of at-risk lesions.	Oral Leukoplakia (T/N)	5 (minimum)	*p* < 0.05
Cervigne et al. [76]	miR-21, miR-181b, and miR-345, and miR-416a	Overexpression of these miRs in OSCC and progressive tissue suggest their involvement in malignant transformation.	Oral Leukoplakia (T/N)	5–9	*p* < 0.001
Hung et al. [77]	miR-31	A significant difference in miR-31 expression was observed between transformed and non-transformed leukoplakia.	Oral Leukoplakia (T/N)	2.25 (mean)	8.34 (HR)
Harrandah et al. [78]	miR-375	Downregulation of miR-375 is associated with malignant transformation in OPLs	Dysplasia, CIS, and verrucous hyperplasia/verrucopapillary hyperkeratosis (T/N)	0.5 (minimum)	*p* < 0.0001

Abbreviations: Follow-Up (F/U), Transformed (T), Non-Transformed (N), Progressing (P), Non-Progressing (NP), hazard ratio (HR), odds ratio (OR).

### 7.2. miRNA and Histone Modification

Besides DNA methylation, the other two epigenetic mechanisms are microRNA (miRNA) expression and histone modification. miRNAs regulate gene expression at the post-transcriptional level by silencing mRNA targets. To date, there are only four longitudinally designed studies looking at the use of miRNAs in predicting the cancer risk of OPLs. No longitudinal study on histone modification in oral precancers has been found. The first miRNA study performed a genome-wide analysis on miRNA expression in leukoplakia with different five-year follow-up outcomes [75]. In this study, leukoplakia lesions included hyperplasia, hyperplasia with hyperkeratosis, and low-grade dysplasia. Following total RNA extraction and qRT-PCR sequencing, those with the most predictive potential were identified using the DESeq Bioconductor package. Four miRNAs, i.e., 208b-3p, 204-5p, 129-2-3p, and 3065-5p, were reported to show a relatively high sensitivity (76.9%) and specificity (73.7%) in differentiating high-risk vs. low-risk lesions. Based on the proposed model that combined the expression of these four miRNAs with age and histological information, 80% of the progressive cases and 63% of the non-progressive cases were correctly predicted. In the second study, global miRNA expression profiles were generated on progressive and non-progressive oral leukoplakia. The results showed that miR-21, miR-181b, miR-345, and miR-416a were found only in progressive leukoplakia and clustered OSCCs but not in non-progressive leukoplakia or normal tissue (*p* < 0.001) [76]. The third study by Hung et al. investigated the potential use of miR-21 and miR31 as markers to predict the progressive outcome of OPLs to OSCC, where OPLs included hyperkeratosis, hyperplasia, and dysplasia [77]. miR-31 was expressed more abundantly in progressive OPLs than in non-progressive OPLs, whereas miR-21 showed no difference between these two OPLs. The sensitivity and specificity for miR-31 were determined to be 87.51% and 73.3%, respectively. This finding indicates that miR-31 may be used as a marker to predict the progression of OPLs to OSCC. The fourth study by Harrandah et al. tested the expression levels of four miRNAs, i.e., miR-7, miR-21, miR-371, and miR-494, in progressive vs. non-progressive OPLs, defined as dysplasia, carcinoma in situ, and verrucous hyperplasia/verrucopapillary hyperkeratosis, in retrospective data with at least six months between dysplasia and malignant diagnosis. They reported that non-progressive OPLs expressed miR-375 at a higher level (*p* = 0.0004) but showed no difference in miR-7, miR-21, or miR-494 expression compared to progressive OPLs [78]. 

## 8. Challenges Facing the Discovery of Predictive Biomarkers

Several major barriers exist that limit progress in finding sensitive and reliable markers for predicting the risk of OPLs progression to OSCC. First, identification and validation of bona fide predictive markers requires either prospectively collected samples or archival samples collected before disease progression and individually followed for long-term outcome in the clinic. Both types of samples are exceedingly rare due to the cost- and time-consuming process of collection. Collecting samples for longitudinal studies with complete follow-up information has been challenging due to the required time, cost, and expertise, compounded by the loss of patient follow-up. As a result, population-based cohorts with long-term and comprehensive follow-up information and samples are either lacking or too small in sample size to achieve the necessary analytical power, especially for whole-genome studies. Second, traditional markers based on protein expression are subject to pitfalls in the methods of detection. Third, the conventional candidate gene or pathway approach has a high likelihood of missing key players currently unknown to affect the disease of interest. Fourth, genetic mutations in OSCCs are wide-spread, irreversible, and sometimes confounded by secondary bystander events. Fifth, the power of genome-/epigenome-wide approaches is highly reliant on technical requirements, such as the quality of data and the analytical power of bioinformatic tools. Sixth, while whole-genome or genome-wide studies offer the advantage of comprehensive new discoveries, it is also difficult to pare down and authenticate all candidates identified by these approaches—a task further complicated by the limited availability of samples for validation. Lastly, although a longitudinal study design is the most ideal model for the discovery and validation of predictive markers, it is not without potential limitations. During the time of follow-up, patients may drop out of the study due to moving or death by other illness or may receive various treatments between sequential biopsies. Those treatments may certainly complicate the analysis by altering the expression of markers and/or the progression outcome of the lesion. This calls into question the significance and validity of markers obtained from samples where some patients received treatment and some did not. Another issue is that the length of time used by different studies to define progressive vs. non-progressive OPLs is somewhat arbitrary. Some studies did not provide sufficient information on follow-up time, and not all studies used the same criteria to define progressive vs. non-progressive cases.

## 9. Future Research Trends

To date, there is no uniform standard of care for patients with diffuse or large-sized low-grade oral dysplasia due to their relatively low malignant transformation rate (4~11%) weighted against the severity of treatment complications. However, for the significant minority that do develop OSCC, the consequence is grave. The visible appearance of OPLs during routine dental visits provides an opportunity to implement risk assessment and cancer-preventative strategies, if only a sensitive and reliable method is available to predict the progressive vs. non-progressive outcome of low-grade OPLs so that unnecessary treatment complications can be avoided. 

Given what has been discussed in this review, several research directions can be envisioned for the future. First and foremost, there is a critical need to establish community-based longitudinal cohorts starting at the early premalignant stage with the collection of detailed clinical information and tissue samples. The second research need is to develop improved whole-genome strategies and bioinformatic tools to perform robust genetic and epigenetic analyses with deep coverage (at a single-nucleotide resolution) to discover a complete spectrum of genetic/epigenetic markers capable of differentiating progressive from non-progressive OPLs. From those markers, novel risk prediction models may be created by selecting the disease-driving ones using cutting-edge analytical tools, such as artificial intelligence and machine learning, for clinical applications. Third is to design large-scale prospective studies to validate identified putative cancer-risk-predictive markers found by longitudinal studies. Finally, it is of fundamental importance to dissect the mechanisms underlying the malignant progression of oral dysplastic lesions, which may reveal druggable targets for cancer prevention even before the appearance of OPLs.

Above all, there is a clinical need to test the identified prediction models on saliva and/or brushing samples, collected from patients with low-grade OPLs, and compare the readouts and their predictive powers to data collected from biopsy samples. Using saliva or brushing cytology samples as surrogates for tissue samples may obviate some practical issues related to multiple biopsies. Several efforts have spearheaded the translational frontiers of salivary diagnostics for OSCC. For instance, by comparing between OSCC and control, several aberrantly expressed cancer-related mRNAs were observed in OSCC saliva from profiling human salivary transcriptome [79]. In addition, salivary proteomic analysis identified elevated levels of salivary proteins compared to normal counterparts. These salivary biomarkers were able to distinguish cancer from benign diseases with high sensitivity and specificity [80,81]. Cytological study of oral cells is also a non-invasive technique that has been harnessed into use for detection of disease progression and therapeutic monitoring. By quantifying the DNA amount from collected cells, an image analysis system was developed to classify cells based on DNA ploidy, which, in term, can triage patients based on the levels of abnormality in their lesions [82,83]. Such a technique was validated for its accuracy at distinguishing at-risk lesions with 100% sensitivity and 86.7% specificity [83]. Longitudinal studies and multicenter screening studies are warranted to disseminate for clinical usage.

In conclusion, there is a critical need for a screening method, preferably non-invasive, to predict the risk of OPLs in becoming OSCCs, which allows early medical interventions that are more effective and less damaging. Finding sensitive and reliable cancer risk-predictive markers for OPLs will not only have a direct impact on OSCC-related health issues but may also have a broad impact on tobacco-related HNSCC. 

## Figures and Tables

**Figure 1 cells-11-01033-f001:**
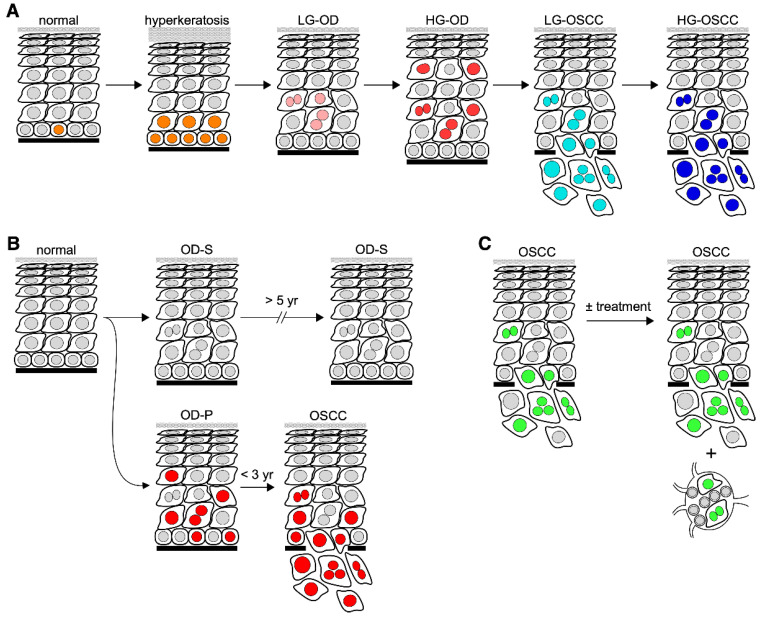
Schematic diagrams of biomarkers in oral carcinogenesis. Diagrams depicting diagnostic markers for differentiating different lesion types (**A**), predictive markers for assessing the risk of oral premalignant lesions in developing cancer (**B**), and prognostic markers for predicting the outcome of oral cancer (**C**). Abbreviations: LG, low-grade; HG, high-grade; OD-P, progressive oral dysplasia; OD-S; static oral dysplasia. Nuclei in (**A**) are labeled with different color schemes to indicate mitotic cells (orange), low-grade dysplastic cells (pink), high-grade dysplastic cells (red), low-grade malignant cells (light blue), and high-grade malignant cells (dark blue). Nuclei in (**B**) are labeled in red to indicate cells expressing high-risk markers for malignant progression. Nuclei in (**C**) are labeled in green to indicate malignant cells.

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
