# Peer review of "On the Cutting Edge of Oral Cancer Prevention: Finding Risk-Predictive Markers in Precancerous Lesions by Longitudinal Studies"

_cells, 2022, doi:10.3390/cells11061033_

Round 1
Reviewer 1 Report
The manuscript is a review on biomarkers that could predict risk of oral premalignant lesions developing into oral squamous cell carcinoma. It is well written and presented and the different approaches to date are clearly described.
Minor comments
- In the Introduction, is risk factor (3) HPV-negative non-smokers really a risk factor??
- Research on saliva and brushing cytology is mentioned in section 9. Please include a brief section describing studies already published using these sample types.
Author Response
Point 1: In the Introduction, is risk factor (3) HPV-negative non-smokers really a risk factor??
Response 1: Revised as suggested (P1, L35-38)
Point 2: Research on saliva and brushing cytology is mentioned in section 9. Please include a brief section describing studies already published using these sample types.
Response 2: A section on saliva and brushing cytology was added (P12, L457-474).
Reviewer 2 Report
This article is a review of precancerous lesions in oral cancer. It outlines the various aspects of the disease and is a useful article. As described in the literature, epithelial dysplasia is an important part of cancerization. In leukoplakia, it is very important for the clinician to know which lesions can be followed up and which ones should be surgically resected.
I understand that immunohistological specimens for changes in CK13 and CK15 would be very useful, but this does not seem to be mentioned in this article. I would be very grateful if you could add this to your paper.
Ref
Kitamura R, et al. Association of cytokeratin 17 expression with differentiation in oral squamous cell carcinoma. J Cancer Res Clin Oncol 2012;138:1299-310.
Author Response
This article is a review of precancerous lesions in oral cancer. It outlines the various aspects of the disease and is a useful article. As described in the literature, epithelial dysplasia is an important part of cancerization. In leukoplakia, it is very important for the clinician to know which lesions can be followed up and which ones should be surgically resected. I understand that immunohistological specimens for changes in CK13 and CK15 would be very useful, but this does not seem to be mentioned in this article. I would be very grateful if you could add this to your paper. Ref : Kitamura R, et al. Association of cytokeratin 17 expression with differentiation in oral squamous cell carcinoma. J Cancer Res Clin Oncol 2012;138:1299-310.
Since the focus of the Kitamura’s paper was on diagnostic markers, the appropriate place for its citation in our paper with a focus on predictive markers from longitudinal studies will be in the background section on biomarkers (P2, L64, Ref. 12).
Reviewer 3 Report
This manuscript summarizes the current state of knowledge regarding search for biomarkers for progression of oral potentially malignant lesions to oral cancer using longitudinal studies. In general, I would say it is well-written and stuck to the subject without deviation. Reading through there are few issues that arose in this reviewer mind arranged in ascending order (top being most important with decreasing order of importance as one moves down):
1. The authors appear to just give a laundry list of all the biomarkers without specifically helping the readers to identify which are more robust than the other. In general, it would have been very helpful to an inquisitive reader if under each subheadings the authors are able to separate markers into those that had robust basis due either to the strength of the methodology employed in their discovery or multiply verified by different research groups and so on, and those that do not have robust basis (weak, or even poor methodology and/or lack of verification). This would help the reader who wants to work in the field to know how to choose which biomarkers are worth further investigations and those which are less worthy.
I know the authors would certainly want to chip in that they discussed the limitations of each method for biomarker discovery under them but reading through them they are just general limitations that apply to the method. I do not believe that each paper cited by the authors is flawless and the method is always robust.
2. There is loss of flow in reading especially from section 5 to section 6 due to many unexplained acronyms. This issue suddenly abated in section 7 as the acronyms used began to be explained. The authors are advised to at least state the meaning of acronyms at least at the first point of use in the early and middle sections of the manuscript.
3. In the introduction, the authors are advised to expunge anything related to oropharyngeal carcinoma. This is a different entity from oral squamous cell carcinoma (OSCC) and usually has so much information of by itself that it should never be lumped with OSCC.
4. Still in the introduction, references should be provided for statement in line 33 ("American Cancer Society 2021" is not enough as a reference, the authors should look for their (ACS's) source or at worse include how to locate the exact webpage where the statement is in the reference), and lines 40-41 (“HPV-negative non-smokers represent a small subset of OSCC patients that 40 are relatively overrepresented by females.”).
By the way, HPV is not an important player in OSCC. Most patients are HPV-negative.
5. In line 43, the authors seem to be a little behind the news about challenges being raised to the present grading of dysplasia. I remember a recent article by Odell et al (PMID: 34418233). The authors should add a few lines regarding challenges and modifications to the present system of grading.
6. I am not really sure whether figure 1 serves any useful purpose apart from also being not esthetically pleasing to me (just personal preference). Left to me, I would ask that it be taken out of the manuscript.
Author Response
Point 1: The authors appear to just give a laundry list of all the biomarkers without specifically helping the readers to identify which are more robust than the other. In general, it would have been very helpful to an inquisitive reader if under each subheadings the authors are able to separate markers into those that had robust basis due either to the strength of the methodology employed in their discovery or multiply verified by different research groups and so on, and those that do not have robust basis (weak, or even poor methodology and/or lack of verification). This would help the reader who wants to work in the field to know how to choose which biomarkers are worth further investigations and those which are less worthy. I know the authors would certainly want to chip in that they discussed the limitations of each method for biomarker discovery under them but reading through them they are just general limitations that apply to the method. I do not believe that each paper cited by the authors is flawless and the method is always robust.
Response 1: Our objective is to provide a synthesized review on a topic critically important but often obscured to the field of cancer risk prediction and prevention, without directly judging the quality of the studies being reviewed. The reviewer’s point is valid and well appreciated, and, to a large part, has been addressed by our summary tables for IHC, genetic, and epigenetic markers, enlisting the key design (sample types, and follow-up time if available) to help inquisitive readers make their own educated judgements. One can also see those verified by different research groups (e.g., p53, p16, PDPN, and MAGE-A in Table 1; several LOHs in Table 2, and p16 methylation in Table 3). To address the point of robustness, we added a new column of reported statistical strength.
Point 2: There is loss of flow in reading especially from section 5 to section 6 due to many unexplained acronyms. This issue suddenly abated in section 7 as the acronyms used began to be explained. The authors are advised to at least state the meaning of acronyms at least at the first point of use in the early and middle sections of the manuscript.
Response 2: Full names for abbreviations based on HUGO gene nomenclature (if available) were provided.
Point 3: In the introduction, the authors are advised to expunge anything related to oropharyngeal carcinoma. This is a different entity from oral squamous cell carcinoma (OSCC) and usually has so much information of by itself that it should never be lumped with OSCC.
Response 3: Revised as suggested (P1, L40-42)
Point 4: Still in the introduction, references should be provided for statement in line 33 ("American Cancer Society 2021" is not enough as a reference, the authors should look for their (ACS's) source or at worse include how to locate the exact webpage where the statement is in the reference), and lines 40-41 (“HPV-negative non-smokers represent a small subset of OSCC patients that 40 are relatively overrepresented by females.”). By the way, HPV is not an important player in OSCC. Most patients are HPV-negative.
Response 4: References for these two statements were provided (P1, L34-35, L42)
Point 5: In line 43, the authors seem to be a little behind the news about challenges being raised to the present grading of dysplasia. I remember a recent article by Odell et al (PMID: 34418233). The authors should add a few lines regarding challenges and modifications to the present system of grading.
Response 5: Please refer to our response to point #3 raised by reviewer #3 (P2, L47-52).
Point 6: I am not really sure whether figure 1 serves any useful purpose apart from also being not esthetically pleasing to me (just personal preference). Left to me, I would ask that it be taken out of the manuscript.
Response 6: We appreciate the comment but prefer to keep the figure for illustration.
Reviewer 4 Report
The manuscript was well-written and organized.
Few comments were listed below:
- In introduction, line 34-36, about three major risk factors:
-Change to “three major groups” is recommended, since HPV-negative non-smokers is not a kind of risk factor.
-Change to (1) oral habit associated, including tobacco, heavy alcohol and betel nut chewing (2) HPV infection (3) no known risk factors are recommended. Although betel nut chewing is more in Southeast Asia, majority of the oral cancer occurred in Southeast Asia.
- In introduction, line 42: mouth floor should be changed to the floor of the mouth
- In introduction, line 46-49 low grade and high grade oral epithelial dysplasia:
-The description about the “low-grade, including mild and moderate dysplasia… high-grade (severe dysplasia)…”is not correct. Based on the WHO consensus (Ref. Oral Diseases 2021; 27:1947-1976, p1954), moderate dysplasia either be splitting into low or high grade or completely classified into high grade dysplasia.
- In Figure 1, the color coding for schematic diagrams was missing. The meaning of changes in different cells was not described clearly. The “hyperkeratosis” should be shown with thickened keratin layer.
Author Response
Point 1: In introduction, line 34-36, about three major risk factors: (a) Change to “three major groups” is recommended, since HPV-negative non-smokers is not a kind of risk factor, and (b) Change to (1) oral habit associated, including tobacco, heavy alcohol and betel nut chewing (2) HPV infection (3) no known risk factors are recommended. Although betel nut chewing is more in Southeast Asia, majority of the oral cancer occurred in Southeast Asia.
Response 1: Revised as suggested (P1, L35-38)
Point 2: In introduction, line 42: mouth floor should be changed to the floor of the mouth
Response 2: Revised as suggested (P1, L42-43)
Point 3: In introduction, line 46-49 low grade and high grade oral epithelial dysplasia: The description about the “low-grade, including mild and moderate dysplasia… high-grade (severe dysplasia)…”is not correct. Based on the WHO consensus (Ref. Oral Diseases 2021; 27:1947-1976, p1954), moderate dysplasia either be splitting into low or high grade or completely classified into high grade dysplasia.
Response 3: We understand that changes to the current 2017 WHO criteria for dysplasia diagnosis have been proposed with emphasis on the architectural features of epithelial dysplasia into a two-tier system, i.e., low-grade and high-grade. However, it is proposed with a caveat that further validation is required. No matter the 2-tier or the conventional 3-tier WHO system, the subjectivity with inter-observer and intra-observer discrepancies is unavoidable. We are currently carefully examining this issue with our longitudinal dysplasia cases with outcome of progression to validate the 2-tier system. To consider the reviewer’s concern, we have modified the statement (P2, L47-52).
Point 4: In Figure 1, the color coding for schematic diagrams was missing. The meaning of changes in different cells was not described clearly. The “hyperkeratosis” should be shown with thickened keratin layer.
Response 4: Revised as suggested (P3)
Round 2
Reviewer 3 Report
The authors appear to have addressed most issues raised in the previous review.